# Food Insecurity and the Association between Perceptions and Trust of Food Advertisements and Consumption of Ultra-Processed Foods among U.S. Parents and Adolescents

**DOI:** 10.3390/nu14091964

**Published:** 2022-05-07

**Authors:** Reah Chiong, Roger Figueroa

**Affiliations:** Division of Nutritional Sciences, College of Human Ecology, Cornell University, Ithaca, NY 14853, USA; rf453@cornell.edu

**Keywords:** food insecurity, food advertisements, ultra-processed foods, dyadic interdependence

## Abstract

Adolescents exposed to food and beverage advertisements (FBAs) typically low in nutrient density can be influenced in their food choices, eating behaviors, and health. This study examines the association between perceptions and trust of FBAs (key predictor) and the outcome of daily consumption of ultra-processed foods (UPFs) in parent-adolescent dyads, with risk of food insecurity as a potential moderator. Cross-sectional data from the Family, Life, Activity, Sun, Health and Eating (FLASHE) study was used to test actor and partner effects using structural equation modeling. The final model was adjusted for parent sex and education level, and effects were compared between dyads at risk of food insecurity (*n* = 605) and dyads not at risk (*n* = 1008). In the unadjusted model, actor effects (parent: b = 0.23, *p* = 0.001; adolescent b = 0.12, *p* = 0.001) and parent-partner effects were found (b = 0.08, *p* = 0.004). The final comparative model produced similar results for dyads not at risk of food insecurity (parent actor: b = 0.27, *p* = 0.001; parent partner: b = 0.10, *p* = 0.01; adolescent actor: b = 0.11, *p* = 0.003). For dyads at risk of food insecurity, only actor effects were significant (parent: b = 0.22, *p* = 0.001; adolescent: b = 0.11, *p* = 0.013). These findings suggest that parents’ favorability towards FBAs influence parent-adolescent unhealthy food consumption, and that this association is different when accounting for risk of food insecurity.

## 1. Introduction

The growing prevalence of childhood obesity continues to be a significant public health concern. Currently, 13.7 million U.S. children and adolescents are affected by overweight and obesity [1]. Suboptimal diets, which include foods low in nutrient density such as ultra-processed foods (UPFs), are one of the main behavioral risk factors associated with obesity [2]. UPFs can be defined as foods with formulations of ingredients made through a series of industrial processes that require sophisticated equipment designed to create convenient (i.e., ready-to-consume), highly profitable products (i.e., inexpensive, non-perishable, highly marketable), and hyper-palatable products prone to compete with fresh foods [3]. Some examples of UPFs include sugar-sweetened beverages (SSB), such as soda and fruit drinks, packaged snacks, and ready-to-heat food products and reconstituted meat products (i.e., hot dogs). In a randomized controlled study of 20 adults, researchers investigated the effects of an ultra-processed diet versus an unprocessed diet on energy intake and weight status after 2-week periods. Hall et al. concluded that participants who were assigned diets with an energy composition mainly derived from UPFs resulted with an increase in energy intake of 500 calories per day, in addition to an increase in body weight and body fat mass [2]. On the other hand, a diet with energy primarily taken from unprocessed foods showed a decrease in these measures despite both diets having matched nutritional parameters [2]. These findings significantly complement the growing body of research that has linked the prevalence of overweight and obesity to the consumption of UPFs in recent years [4,5,6]. However, Poti, Braga, and Qin’s narrative review on this association has suggested that further studies with stronger designs are needed to differentiate impacts on metabolic outcomes between the method in which the food is made (processing) and the food’s nutritional value [7].

In the U.S., UPFs contribute to over half of the population’s energy intake [8], and has increased over time [9]. Previous studies aimed at adolescent populations have found associations between food advertisements and the increased intake of foods lower in nutrient density [10], such as SSB [11]. In a study by Thai and colleagues [10], adolescents were more likely to consume more of these types of foods and drinks (candies, SSB, potato chips, etc.) if they positively perceived and trusted food advertisements. UPFs, the key outcome in this study, are also low in nutrient density, and have additional implications because they highlight the ways that food is processed. Overall, the impact of food and beverage advertisements (FBAs) on the consumption of UPFs specifically is limited. Furthermore, since UPF consumption is higher among those who have low income [9], it is important to also investigate how food insecurity, or the inconsistent access to sufficient food [12] may disproportionally affect these households when assessing the role of FBAs on the consumption of UPFs.

The targeted marketing and positive perceptions and trust towards advertisements promoting foods low in nutrient density such as UPFs among adolescents may continue to reinforce choice and consumption of these types of foods. Not only do FBAs heavily target youth (i.e., through television advertising, product packaging, endorsements), but they directly influence youth food preferences, dietary attitudes, eating behaviors, and consequently, health outcomes [13]. Compared to adults, children and adolescents are more susceptible to branding and food marketing messages conveyed through FBAs [14]. Among U.S. youth, however, adolescents (aged 12–16) are largely more exposed to these advertisements across different outlets (i.e., television, social media) compared to younger children [15]. As the access and use of smartphones, internet, and media increased in recent years amongst U.S. adolescents [16], food marketers have also shifted towards the use of these newer media platforms to communicate, engage, and shape consumer behavior among this age group [17]. High exposure and receptivity to FBAs are associated to adolescent food choices, unhealthy eating behaviors, and weight-related outcomes [18,19,20,21]. Most importantly, a majority of adolescent-targeted FBAs are low in nutrient density (high in fat, sodium, and sugar) and include fast food, sweets, beverages (specifically fruit drinks and soft drinks), and snacks [22].

Furthermore, recent studies have elucidated on family interdependence and the role that parents play in their children’s dietary and health behaviors [23]. According to Scaglioni et al., the family environment determines what food is available for a child or adolescent to consume, and what foods they will try. They also establish dietary habits, such as healthy eating and food preparation. Parents are powerful role models in the family unit, especially during the key developmental stage of adolescence. Ultimately, they shape their children’s patterns and behaviors later on in life [24,25], and help develop their eating routines from a young age [26]. Research, however, remains limited on the influence of FBAs and UPFs consumption in the familial context and on parent-adolescent relationships. As such, it is central to our inquiry to consider how parents’ own favorability towards FBAs and their consumption of UPFs is associated with their adolescent’s own favorability towards FBAs and consumption of UPFs, especially for family units with socioeconomic hardships (i.e., at risk of food insecurity).

To address these gaps, the goal of this study is to investigate the perceptions and trust of FBAs on daily consumption of UPFs in parent-adolescent dyads via three aims. The first aim is to examine actor effects of FBAs and daily consumption of UPFs in both parents and adolescents. In other words, the study hypothesizes that parents and adolescents who are more favorable towards FBAs will consume more UPFs daily. The second aim is to examine partner effects of dyadic pairs. It has been suggested that a bidirectional relationship could exist, and that children may also influence their parents eating behaviors [27]; however, we hypothesize that adolescent favorability towards FBAs will not influence parent consumption of UPFs based on the functions of typical family structures and the influence that parents have on their children. The final aim is to examine whether food insecurity acts as a moderator between the association of perception and trust of FBAs and daily servings of UPFs. We hypothesize that the actor and partner effects will vary for dyads who are at risk of food insecurity, and dyads who are not.

## 2. Materials and Methods

### 2.1. Data Source

The current study adopts a cross-sectional design and draws data from the Family Life, Activity, Sun, Health, and Eating (FLASHE) study. Funded by the National Cancer Institute to collect information on cancer-preventive behaviors, the parent study recruited a demographically representative sample from the Ipsos’ Consumer Opinion Panel, and enrolled a total of 1945 parent or caregiver and adolescent (ages 12–17) dyads between April 2014 and October 2014. Dyads were then randomly assigned to two groups. The first group was a survey-only group where dyads completed an online survey mainly focused on diet and physical activity behaviors and correlates. In addition to completing the same online survey, the second group of parent-adolescent dyads were provided with accelerometers for adolescents to wear. Both groups also completed a demographic questionnaire. Only measures from the diet survey were used for this study. The main study constructs in this study drawn from FLASHE include advertising/media perception, food consumption, risk of food insecurity, and demographic data. A listwise deletion approach was used to address missing data.

### 2.2. Study Measures

#### 2.2.1. Consumption of UPFs

Total daily consumption of UPFs was assessed using the Dietary Screener Questionnaire (DSQ) [28]. Parent and adolescent participants reported the weekly frequency of consumption on the following foods: sweetened fruit drinks, soda, energy drinks, fried potatoes, tacos, heat and serve, processed meat, hamburgers, fried chicken, candy/chocolate, cookies and cakes, and frozen desserts. In total, 12 dietary indicators were chosen by researchers based on the NOVA classification system of UPFs [29]. The response scale ranged from 1 (‘I did not consume [food or beverage] during the past week’) to 6 (‘I consumed [food or beverage] 3 or more times per day during the past week’). In order to develop the composite score, the response scale was recoded to denote daily servings of UPFs, ranging from 0 to 3 daily servings on each indicator. The total amount of servings per day was used as the dependent variable denoting total daily consumption of UPFs.

#### 2.2.2. Perception and Trust of FBAs

Perception and trust of FBAs were measured using three 5-point Likert-type items in both parents and adolescents. Individually, participants were asked to think about marketing messages heard or seen through various print, auditory, and digital platforms. Parents and adolescents were prompted with the following statement, “When I see advertisements for food or drinks…” that connected to each of the following connecting statements: (a) “I want to try the advertised foods or drinks,” (b) “I think the advertised foods or drinks will taste good,” and (c) “I trust the messages advertised.” Participants provided a response to each item, ranging from strongly disagree (1) to strongly agree (5). The average score across the three items was used as an independent variable denoting perception and trust of FBAs. In all models, a measurement model was specified for perceptions and trust of FBAs. The Cronbach’s Alpha value for the predictor of perceptions and trust of FBAs is for parents is 0.83 and 0.84 for adolescents. There was also moderate to strong positive cross-factor correlations for both parents (r = 0.55–0.76) and adolescents (r = 0.56–0.76). Since perceptions and trust of FBAs is complex and not directly observable, it was denoted as an exogenous latent variable measured by the 3 survey items as its observed indicators for both parents and adolescents. Residual correlations between the same parent and adolescent items were established to account for factors not depicted in the model [30]. The characterization of perception and trust follows a previous study that also used the same data set and measurements [10].

#### 2.2.3. Risk of Food Insecurity

Food security was measured using two items [31]. Parents were asked to rate how true the following statements were for them and their households in the past year: (a) “We worried whether our food would run out before we got money to buy more,” and (b) “The food that we bought just didn’t last, and we didn’t have money to get more.” Response options included Never True (0), Sometimes True (1), and Often True (2). For either of the statements, values were set to 0 for participants who responded Never True while values for participants who responded Sometimes True or Often True were set to 1. Finally, these values were added together to create a score that captures risk of food insecurity. Participants who had a sum of 0 were labeled as not at risk for experiencing food insecurity, while participants who had a sum of 1 or 2 were labeled at risk for experiencing food insecurity.

#### 2.2.4. Covariates

The following variables were considered covariates: sex, age, race, education level, household income, and receipt of food assistance. Parents and adolescents were asked whether they identified as male or female, and to fill in their age. The 4 response options for race were Hispanic, Non-Hispanic Black or African American only, Non-Hispanic White only, or Non-Hispanic. Parents were asked to indicate their education level with 4 response options ranging from less than a high school degree, a high school degree or GED, some college but not a college degree, or a 4-year college degree or higher. Parents were also asked to indicate their combined annual income in their households with 9 response options ranging from $0 to $9999 to $200,000 or more. Lastly, parents were asked to indicate whether they received food assistance. In the final model, only parent sex and parent education level were considered.

### 2.3. Analytic Plan

Descriptive statistics were initially completed to summarize information on parents’ and adolescents’ demographics and key study measures (in the form of average scores with standard deviations). Inter-item correlation (Pearson’s r) and internal consistency (Cronbach’s alpha) between predictor, moderating, and response variables among complete pairs were also measured as a preliminary assessment of associations. In order to test dyadic actor effects (aim 1) and partner effects (aim 2) of perceptions and trust of FBAs and consumption of UPFs, actor–partner interdependence model within a structural equation modeling (SEM) framework was used [32,33]. A multigroup approach was used to compare actor and partner effects between two groups (aim 3) while adjusting for parents’ sex and education level: (1) parents who were at risk of food insecurity, and (2) parents who were not at risk of food insecurity. All values presented are standardized. Model fit, specifically chi-square difference (Δχ^2^), RMSEA, CFI, SRMR, and AIC, were also assessed [34]. Stata and R statistical software (version 14) was used to conduct analyses.

## 3. Results

### 3.1. Demographics and Key Measures

A total of 1859 dyads were included in this study, but only 1613 were accounted for in the final model (Figure 1). Among the parent group, a majority were female, non-Hispanic White, and were middle-aged. Most parents attended a college or received a college degree and indicated that they did not receive food assistance; most were not at risk of food insecurity. Table 1 describes the demographic findings in greater detail for both parent and adolescent groups. Descriptive statistics were also conducted on key measures. Overall, parents consumed an average of 3.16 ± 3.18 servings of UPFs daily, while adolescents consumed 4.05 ± 3.80. For the predictor variable of perception and trust of FBAs, the average total score for the parent group was 8.63 ± 3.41 while the average total score for adolescents was 8.97 ± 4.04. When accounting for risk of food insecurity, both daily servings of UPFs and scores of perception and trust of FBAs were higher among parents and adolescents who were at risk compared to those not at risk. A summary of descriptive findings according to risk of food insecurity can be found in Table 2.

Several significant positive correlations were found between and across all variables of interest. The most important to note is a moderate positive correlation between parent and adolescent predictor variables as well as parent and adolescent outcome variables. This implies that increases in total daily consumption of UPFs in one group is correlated with increases in the same variable for the other group, r = 0.59 (*p* = 0.01). Furthermore, an increase in perceptions and trust of FBAs in the parent group is also correlated with an increase in the same variable for adolescents, r = 0.51 (*p* = 0.01). It seems that parents’ perceptions and trust of FBAs is also positively correlated with the outcome variable in the parent group, r = 0.35 (*p* = 0.03) as well as the adolescent group, r = 0.26 (*p* = 0.04). On the other hand, adolescents’ perceptions and trust of FBAs was only significantly correlated with adolescents’ consumption of UPFs, r = 0.38 (*p* = 0.02), but not parents’ consumption of UPFs, r = 0.16 (*p* = 0.16). Lastly, both items denoting risk of food insecurity had a strong positive correlation with one another, r = 0.76 (*p* = 0.01). These correlations are listed in Table 3.

### 3.2. Measurement Model

Two latent variables were developed consisting of 3 observable indicators to reflect the concept of perceptions and trust of FBAs (Figure 2).

Due to potential and unknown family influences on responses, residual errors of parent indicators were set to correlate with corresponding adolescent indicators. Overall, the measurement model indicated good fit. Standardized factor loadings and its corresponding R^2^ values indicate that each item adequately represents perceptions and trust of FBAs for both parents and adolescents. The first two indicators for each model (“I want to try the advertised foods or drinks” and “I think the advertised foods or drinks will taste good”), however, are more reflective of convergent validity than the last indicator (“I trust the messages advertised”). Nevertheless, we proceeded to use this measurement model in the structural equation model.

### 3.3. Unadjusted Structural Equation Model of Actor (AIM 1) and Partner (AIM 2) Effects

The outcome of consumption of UPFs was set as the endogenous variable in the structural equation model, while the predictor of perception and trust of FBAs remained as a latent variable for both parents and adolescents (Figure 3). The assessment of model fit indicated the model was satisfactory.

Significant coefficients were found between parents’ perceptions and trust of FBAs and their own consumption of UPFs (β = 0.236, *p* < 0.001) as well as their adolescents UPF consumption (β = 0.087, *p* < 0.004). Adolescents’ perceptions and trust of FBAs was also associated with their own consumption of UPFs (β = 0.120, *p* < 0.000); however, it did not significantly relate to the consumption of UPFs in parents (β = 0.007, *p* < 0.814), as predicted by our hypothesis.

### 3.4. Actor–Partner Interdependence Model Accounting for Risk of Food Insecurity (AIM 3)

Finally, the structural equation model was tested to include a moderator representing food insecurity. Actor and partner effects were compared between dyads at risk of food insecurity and dyads not at risk of food insecurity (Figure 4), while adjusting for sex and education level of parents.

Overall, model fit indices suggest a good fit. For dyads not at risk of food insecurity (*n* = 1008), findings were similar with findings for Aim 1 and 2. Parents perception and trust of FBAs were significantly associated with consumption of UPFs for both themselves (β = 0.271, *p* < 0.00) and their adolescents (β = 0.100, *p* < 0.01). Perception and trust of FBAs for adolescents was also significantly associated with their own UPF consumption (β = 0.115, *p* < 0.003); however, partner effects for adolescents were not significant (β = −0.018, *p* < 0.639) for this group. On the other hand, only actor effects for both parents (β = 0.227, *p* < 0.000) and adolescents (β = 0.118, *p* < 0.013) were significant for dyads at risk of food insecurity (*n* = 608). As a whole, the actor and partner relationship between perception and trust of FBAs and UPF consumption in parent-adolescent dyads are different according to food insecurity risk.

## 4. Discussion

The goal of this study was to examine associations between perceptions and trust of FBAs and consumption of UPFs among parent-adolescent dyads using APIM, and the role that risk of food insecurity has in moderating this association. In unadjusted models, we found significant actor effects for both parents and adolescents, and significant parent partner effects. Dyads who were not at risk of food insecurity mimicked these findings; however, only actor effects for both parents and adolescents were significant among dyads who were at risk of food insecurity. Overall, these findings suggest that favorability towards FBAs is linked to suboptimal dietary behaviors, parents’ own perceptions of FBAs is also linked to what their adolescents consume, and that these associations are independent of the dyad’s risk of food insecurity.

In sum, perceptions and trust of FBAs is significantly associated with consumption of UPFs among parents and adolescents. Parental perceptions and trust of FBAs were also significantly associated with adolescents’ consumption of UPFs. As such, there was partial support for our first two hypotheses. These findings highlight a degree of vulnerability for U.S. adolescents as targets of food marketing [35] since their consumption of foods low in nutrient density is not only associated with their own media perceptions but also their parents’ perceptions as well. Parents could act as mediators by regulating FBA exposure to their children [36]; thus, acting on their own favorability towards food advertisements could also have implications on their children’s behaviors. Our findings support many previous studies that have identified the important role that parents play in shaping the home food environment and their adolescents’ eating behaviors [37,38]. One dyadic study even identified that greater mobile media use among parents influences their children’s consumption of foods low in nutrient density [39].

Even though our findings indicate that parents and adolescents who are at risk of food insecurity have higher average scores of perceptions and trust of FBAs compared to dyads not at risk, there may be contextual factors that explain the lack of significant partner effects. Adolescents who experience food insecurity are aware of their family hardship even though parents may not be open to discuss it [40]. They assume adult responsibilities, such as finding a job to help provide for their families’ needs and for themselves [41,42]. Even though youth from socio-economically disadvantaged backgrounds and belonging to minority groups experience higher exposure to food advertisements low in nutrient density [43,44], adolescents’ own food consumption may not be swayed by their parents’ favorability towards FBAs; if they were aware of their family’s financial stress, purchasing items could lead to a depletion of their family’s resources and potentially their own earned money [41]. While testing the study’s third aim, we expected that dyads who were at risk of food insecurity would have significantly greater dyadic actor and partner associations between favorability of FBAs and consumption of UPFs compared to dyads without risk of food insecurity. In descriptive analysis, dyads who were at risk of food insecurity had higher average daily servings of UPFs than dyads not at risk of food insecurity. Several studies have noted that food advertisements are more prevalent in low-income, minority neighborhoods [43,45,46]. Since income is a major determinant to food security [47], our findings suggest that families who experience food insecurity are not just vulnerable to food advertisements, but also to the consumption of UPFs [48].

Among dyads not at risk of food insecurity, findings mirror findings from unadjusted models in that greater favorability towards FBAs among parents is associated with not only their own consumption of UPFs, but also adolescents’ UPFs consumption as well. For families who were at risk of food insecurity, actor effects, but not partner effects, were statistically significant. Regardless of food insecurity risk, there may be additional environmental (physical and social) factors that contribute to this association between favorability of FBAs and consumption of UPFs among U.S. parents and adolescents. For example, in addition to the physical and digital environment of food marketing that adolescents are exposed to and greatly engage with [49,50,51], endorsements from celebrities and adolescents’ own peers could potentially influence receptivity of FBAs low in nutrient density [52,53] and food consumption [54,55] alongside parents. Overall, more research is needed to understand the environmental and social context behind these significant actor- and parent-driven effects. Gathering information on the mechanisms and context of these associations could inform future intervention work among this population. One example would be the development of a nutrition education curriculum that effectively improves media literacy of FBAs among families, which has shown to facilitate discussion of food marketing messages and also improve fruit and vegetable consumption in youth [56]. Findings from the current study and further research could also lead to the design of a social marketing campaign that empowers adolescents to engage in peer discussion, think critically about the messages they receive from FBAs low in nutrient density, and advocate for change in their communities.

This study has a few strengths. This study analyzes data from a demographically representative sample of U.S. parents and adolescents, which provide strong support towards generalizability. In addition, the operationalization of the study’s independent variable (favorability of FBAs) enabled the reduction in data dimensionality. This means that instead of aggregating indicators to denote an underlying concept as a composite score, we used a latent variable comprised of observed indicators all while minimizing measurement error. Lastly, the study’s APIM approach accounts for the clustered nature of the parent-adolescent dyadic data as a unit. In other words, such an approach allows for capturing the interdependence of parent–adolescent relationships in the family context, rather than examining associations among key outcomes individually for each family member.

It is worth noting that this study has several limitations. Although the actor–partner interdependence varies between groups overall, which paths are significantly different with one another is unclear. For example, the findings of this study do not determine whether or not the significant association between favorability towards FBAs in parents and their own UPF consumption is significantly different between groups. In order to confirm these differences between paths, additional analyses with model constraints are necessary. Furthermore, the APIM model only adjusted for two covariates: parents’ sex and parents’ education level. Accounting for other sociodemographic variables such as income, race, ethnicity, in addition to other key covariates, could have led to different findings since they are also linked to poor dietary intake. Lastly, the FLASHE was cross-sectional which indicates that causal inferences cannot be made from this study. Longitudinal studies are warranted to examine the causal relationship between food advertisements and consumption of unhealthy foods in U.S. families.

## 5. Conclusions

When examining associations between perceptions and trust of FBAs and consumption of UPFs in parent-adolescent dyads, actor effects were significant regardless of food insecurity risk; parent-adolescent partner effects were also significant, but only for dyads not at risk of food insecurity. The mechanisms behind these associations are unclear, and more studies are needed to explore the content and impact of FBAs, variation in dyadic interdependence, and ultimately how these affect the connection between consumption of foods low in nutrient density and diet-related chronic disease outcomes.

## Figures and Tables

**Figure 1 nutrients-14-01964-f001:**
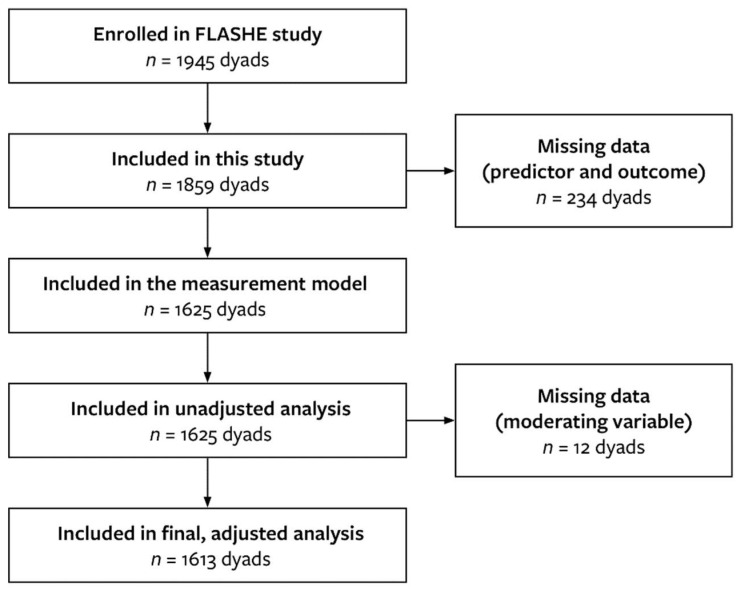
Flow diagram showcasing sample sizes for analysis stages. Listwise deletion was used to take care of missing data.

**Figure 2 nutrients-14-01964-f002:**
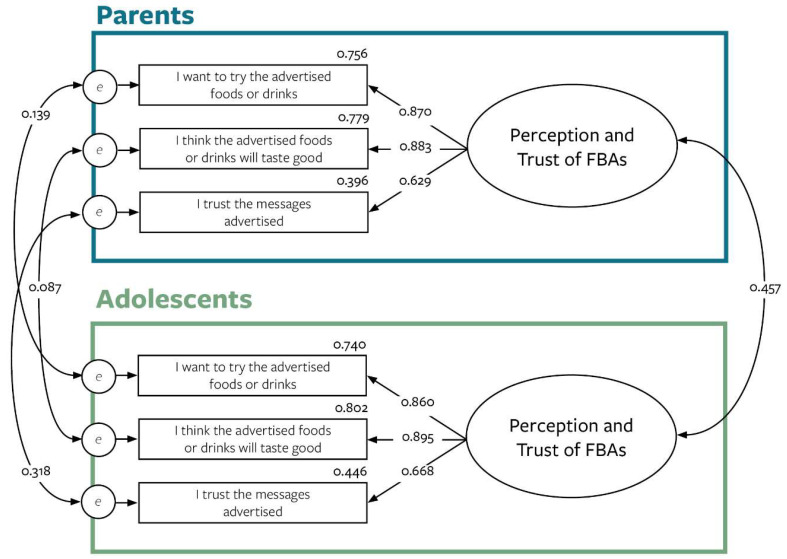
Perceptions and trust of FBAs set as latent variables. Variables in the blue box reflect the measurement model for parents while variables within the green box reflect the model for adolescents. R^2^ values are provided above each indicator. Key: rectangles = indicators; ellipses = latent variables; small circles = residual errors; curved double-ended arrows = covariances; straight single-ended arrows = directional paths.

**Figure 3 nutrients-14-01964-f003:**
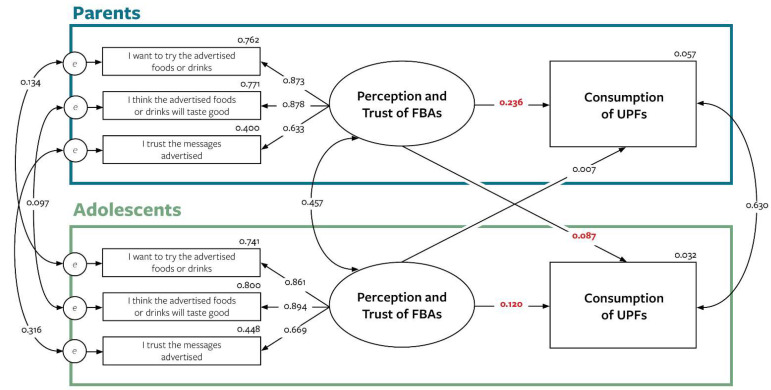
Actor and partner effects of perception and trust of FBAs and consumption of UPFs in parent-adolescent dyads. Significant regression coefficients are bolded in red. Model fit: Δχ^2^ = 49.518, df = 13, *p* = 0.001; RMSEA = 0.042 (90% CI = 0.030 to 0.054), *p* = 0.85; CFI = 0.994; SRMR = 0.026; AIC = 39525.613.

**Figure 4 nutrients-14-01964-f004:**
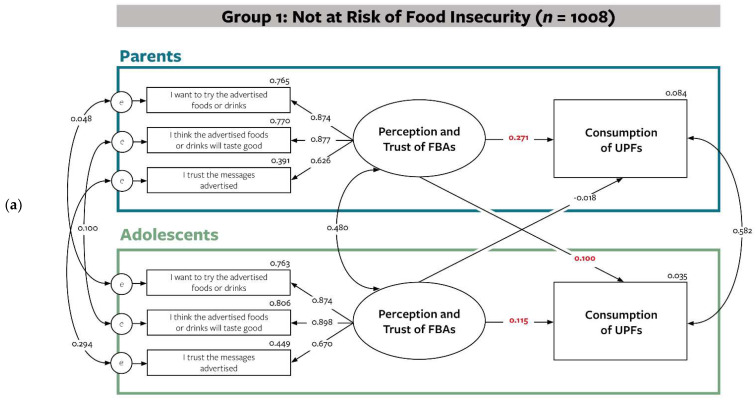
Actor and partner effects of perception and trust of FBAs and consumption of UPFs in parent-adolescent dyads accounting for food insecurity. Significant regression coefficients are bolded in red. Both models adjusted for parents’ sex and parents’ education level. (**a**) SEM model results for dyads not at risk of food insecurity; (**b**) SEM model results for dyads at risk of food insecurity. Model fit: Δχ^2^ = 59.794 & 45.814, df = 26, *p* = 0.001; RMSEA = 0.037 (90% CI = 0.027 to 0.047), *p* = 0.98; CFI = 0.990; SRMR = 0.036; AIC = 44485.329.

**Table 1 nutrients-14-01964-t001:** Descriptive statistics of demographic variables of parent and adolescent dyads.

Demographic Variable	Parents—1859 (%)	Adolescents—1859 (%)
**Sex**		
Female	1325 (74)	843 (50)
Male	468 (26)	835 (50)
NA	66	181
**Race**		
Hispanic	130 (7)	168 (10)
Non-Hispanic Black or African American only	314 (18)	283 (17)
Non-Hispanic White only	1229 (69)	1061 (64)
Non-Hispanic Other	105 (6)	154 (09)
NA	81	193
**Age**	18–34 years 202 (11) 35–44 years 781 (44) 45–59 years 758 (42)60+ years 52 (3) NA 66	12 years old 224 (13) 13 years old 336 (20) 14 years old 280 (17) 15 years old 305 (18) 16 years old 331 (20) 17 years old 206 (12) NA 177
**Level of Education**	Less than a high school degree 22 (1)	-
	A high school degree or GED 301 (17)	
	Some college but not a college degree 634 (35)	
	A 4-year college degree or higher 830 (46)	
	NA 72	
**Household Income**	$0 to $99,999 1406 (79)	-
	$100,000 or more 366 (20)	
	NA 87	
**Food Assistance Participation**	Yes 308 (17)	-
	No 1420 (82)	
	I don’t know 3 (0)	
	NA 128	
**At Risk of Food Insecurity**	Yes 666 (36)	
	No 1193 (64)	-

Note: As a result of missingness, the sample frequencies may not add to 100%.

**Table 2 nutrients-14-01964-t002:** Descriptive statistics of outcome, predictor, and moderating variables according to risk of food insecurity.

	Group 1 Not at Risk of Food Insecurity		Group 2 At Risk of Food Insecurity	
**Key Study Variables**	Parents Mean ± SD (*n*)	Adolescents Mean ± SD (*n*)	Parents Mean ± SD (*n*)	Adolescents Mean ± SD (*n*)
Outcome: daily servings of ultra-processed food indicators				
Sweetened food drinks	0.23 ± 0.45 (1074)	0.41 ± 0.53 (1028)	0.40 ± 0.70 (664)	0.60 ± 0.73 (616)
Soda	0.39 ± 0.63 (1071)	0.45 ± 0.58 (1025)	0.58 ± 0.85 (660)	0.55 ± 0.71 (613)
Energy drinks	0.04 ± 0.19 (1063)	0.06 ± 0.23 (1021)	0.13 ± 0.41 (657)	0.13 ± 0.39 (607)
Fried potatoes	0.23 ± 0.27 (1072)	0.32 ± 0.32 (1030)	0.30 ± 0.38 (659)	0.38 ± 0.42 (613)
Tacos, burritos, and similar dishes	0.16 ± 0.19 (1070)	0.20 ± 0.23 (1029)	0.20 ± 0.32 (659)	0.27 ± 0.41 (616)
Heat & serve	0.13 ± 0.23 (1072)	0.24 ± 0.32 (1036)	0.20 ± 0.38 (663)	0.30 ± 0.45 (616)
Processed meat	0.24 ± 0.27 (1071)	0.30 ± 0.35 (1032)	0.31 ± 0.38 (662)	0.38 ± 0.46 (609)
Hamburgers	0.18 ± 0.19 (1077)	0.22 ± 0.24 (1035)	0.24 ± 0.33 (666)	0.30 ± 0.38 (618)
Fried chicken	0.12 ± 0.17 (1069)	0.20 ± 0.27 (1031)	0.20 ± 0.36 (660)	0.26 ± 0.39 (611)
Candy/chocolate	0.43 ± 0.46 (1074)	0.51 ± 0.51 (1030)	0.39 ± 0.48 (659)	0.50 ± 0.57 (611)
Cookies, cakes, and similar treats	0.29 ± 0.35 (1072)	0.41 ± 0.43 (1026)	0.33 ± 0.44 (656)	0.42 ± 0.49 (611)
Frozen desserts	0.21 ± 0.26 (1074)	0.29 ± 0.31 (1029)	0.22 ± 0.34 (662)	0.31 ± 0.42 (618)
Potato chips	0.27 ± 0.29 (1068)	0.37 ± 0.38 (1032)	0.32 ± 0.38 (664)	0.45 ± 0.46 (612)
Sugary cereals	0.11 ± 0.21 (1074)	0.25 ± 0.35 (1035)	0.18 ± 0.38 (664)	0.33 ± 0.46 (617)
**Total Daily Consumption**	2.72 ± 2.38	3.66 ± 3.11	3.96 ± 4.14	4.77 ± 4.70
Predictor: perception and trust of food & beverage advertisements “When I see advertisements for foods or drinks…”				
I want to try the advertised foods or drinks	2.85 ± 1.39 (1193)	3.03 ± 1.51 (1193)	3.35 ± 1.00 (666)	3.38 ± 1.34 (666)
I think the advertised foods or drinks will taste good	3.03 ± 1.36 (1193)	3.12 ± 1.52 (1193)	3.41 ± 0.96 (666)	3.48 ± 1.30 (666)
I trust the messages advertised	2.33 ± 1.25 (1193)	2.46 ± 1.41 (1193)	2.62 ± 1.05 (666)	2.77 ± 1.34 (666)
**Average Total Scores**	8.20 ± 3.72	8.60 ± 4.19	9.38 ± 2.61	9.64 ± 3.67

Note: As a result of missingness, the sample frequencies may not add to 100%.

**Table 3 nutrients-14-01964-t003:** Pairwise correlation matrix of variables of interest.

Key Variables	1	2	3a	3b	4	5
1. Total daily consumption of UPFs	-					
2. Perceptions and trust of FBAs	0.35 *	-				
3a. We worried whether our food would run out before we got money to buy more	0.15	0.05	-			
3b. The food that we bought just didn’t last, and we didn’t have money to get more	0.16	0.07	0.76 ***	-		
4. Total daily consumption of UPFs	0.59 ***	0.26 *	0.10	0.11	-	
5. Perceptions and trust of FBAs	0.16	0.51 ***	0.03	0.05	0.38 *	-

Note: Coefficients with asterisks are significant (* *p* < 0.05, *** *p* < 0.001). Variables in blue represent parents’ responses while variables in green represent adolescents’ responses. 3a & 3b refer to food insecurity items. Key: UPFs = ultra-processed foods, FBAs = food and beverage advertisements.

## Data Availability

Data are available in a publicly accessible repository that does not issue DOIs. Publicly available datasets were analyzed in this study. This data can be found here: https://cancercontrol.cancer.gov/brp/hbrb/flashe-study/flashe-files accessed on 31 March 2022.

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
