# Peer review of "Food Insecurity and the Association between Perceptions and Trust of Food Advertisements and Consumption of Ultra-Processed Foods among U.S. Parents and Adolescents"

_nutrients, 2022, doi:10.3390/nu14091964_

Round 1

Reviewer 1 Report

Thank you for inviting me to review this manuscript. Authors tried to examines the association between perceptions and trust of FBAs (key predictor) and the outcome of daily consumption of ultra-processed foods (UPFs) in parent-adolescent dyads with risk of food insecurity as a potential moderator. Authors have done a great job. I leave some indications that could improve the manuscript. 

  • Please include a flow chart indicating the final sample of the analysis and the reasons for exclusion.
  • Table 1 should go at the end of the paragraph.
  • The practical applications of the study could be slightly increased, highlighting its importance.
  • To me, Table 4 can be deleted. It is sufficient to indicate this information in the text.  

Best wishes,

Reviewer 2 Report

Overall comment: ads and advertisement are used interchangeably. It would be preferable to pick one and maintain the consistency

Line 34: it would be helpful to have examples of what are part of the SSB category. Perhaps, some of the items listed in line 122 could be included in line 34.

Line 53: Missing “of” in “consume more OF these types of foods and drinks”

Line 59: It would be important to define what food insecurity is

Line 76: all types of beverages or SSB?

Line 122-124: Is there a specific reason the listed food and beverage items are in brackets?

Line 187: “chi-square difference” seems to be in a different font size

Line 188: Was there a specific reason why two different statistical packages were used to run the analyses?

Line 191: It would be helpful to provide a quick explanation for the missing participants (e.g., incomplete data)

Line 196: the sentence seems to be cut short

Line 200: there seems to be a misalignment at the start of that line

Table 2: Are the values for adolescents in group 1 and group 2 under the “Perception and trust of food advertisements” correct? The same exact values are listed for both groups

Table 3: The font in the “key variables” column seems to be different to the rest of the text. I would suggest being constant throughout. Also, do you need to bold the text to highlight statistically significant variables? It seems redundant since you are also providing asterisks for statistically significant associations

Overall: what is the difference between “foods low in nutrient quality” “foods higher in energy density” “low in nutrient density” and “non-nutrient dense foods.” Are these all referring to the same types of foods? It would be best to select one throughout the text.

Line 315: “minoritized” sounds a bit odd. Perhaps “minority” groups is a better fit
